# The Labour Share, Government Expenditure and Income Inequality of Post-Soviet Countries

**Bruno S. Sergi** [1,*] **, Svetlana Balashova** [2] **and Svetlana Ratner** [2,3]

1 Department of Economics, University of Messina, Piazza Pugliatti, 198122 Messina, Italy
2 Department of Economic and Mathematical Modelling, Faculty of Economics, Peoples' University of Russia (RUDN University), Miluh-Maklaya, 6, 117519 Moscow, Russia; balashova-sal@rudn.ru (S.B.); ratner_sv@rudn.ru (S.R.)
3 V.A. Trapeznikov Institute of Control Sciences, Russian Academy of Sciences, Profsoyznaya, 65, 117997 Moscow, Russia
* Correspondence: bsergi@unime.it

**Abstract:** This study analyses the influence of economic growth on inequality, concentrating on the role of governments as mediators. The period studied is from 2000 to 2020, encompassing 11 post-Soviet countries. The primary estimation method used is the two-stage least squares for panel data. Despite the differences in the economic and political systems at the current development stage, the post-Soviet countries share a common pattern in terms of the relationship between economic growth, the labour income share and the level of inequality, which we first show in this article. Government expenditure has the potential to reduce inequality. However, its effectiveness depends largely on government efficiency and the development of democratic institutions. Despite the increase in government spending on education, more is needed to reduce income inequality. Increased economic performance, productivity, and high-quality state institutions are necessary for this change.

**Keywords:** labour share; income inequality; post-Soviet countries; government expenditures; income distribution; TSLS

## 1. Introduction

The United Nations recognises inequality within and between countries as one of the most critical barriers to sustainable development. The widening inequality has significant consequences for economic growth and macroeconomic stability. The concentration of political and decision-making power can result in suboptimal utilisation of human resources and political and economic instability, which decrease investments and raise the risk of crises. In addition, higher levels of inequality reduce economic growth by limiting the ability of lower-income households to maintain their health and accrue physical and human capital (Corak 2013; Autor and Dorn 2013). Entrenched disparities in outcomes can markedly compromise individuals' academic and professional decisions. Despite some signs of progress in reducing inequality, wide differences in wealth inequality remain between countries (Brzezinski and Sałach 2021).

There is an extensive body of literature regarding the link between inequality and economic growth. Numerous studies indicate a direct or inverse correlation between inequality and economic progress. The state acts as a mediator in the relationship between inequality and economic growth. The share of labour in the GDP is crucial for the distribution of income. A decrease in the wage share leads to an increase in the share of capital in the national income and growing inequality due to the concentration of capital, as Thomas Piketty argues in his famous book Capital in the Twenty-First Century (Piketty 2014). However, research has shown that this relationship is not straightforward.

Most of the existing literature in this area has focused on advanced countries. Recently, there has been growing interest in the problem of inequality in developing countries (Ng

et al. 2019; van Treeck and Wacker 2020; Tian et al. 2022). However, only a few researchers have addressed the problem of inequality in post-Soviet countries, which still have some common inherited features of economic development, such as market imperfections, resource misallocation, corruption, and nepotism.

Significant political and economic reforms took place following the dissolution of the Soviet Union and the formation of 15 new states. The abrupt liberalisation of the economy and the accompanying market system led to a sharp increase in inequality in the new states in the early 1990s. Even though, since 2000, there has been a significant reduction in income inequality in most states, inequality remains the fundamental problem of social and economic development in the post-Soviet countries.

This paper aims to determine the precise impact of the labour share in the GDP on the level of income inequality in the former Soviet countries, considering the level of economic growth. The former Soviet countries represent a unique case of countries with a common socialist past but a different capitalist present and implementation of different types of capitalism. The study's primary hypothesis suggests that the complete set of post-Soviet countries can be divided into several clusters characterised by different models of capitalism and, as a consequence, different patterns of state involvement in the fight against inequality. Therefore, the central research question we address is to identify the most successful model for reducing income inequality. To the best of our knowledge, this is the first time that this group of countries has been considered from the perspective of the state's mediating role between economic growth and inequality.

Panel data on inequality in post-Soviet countries since 2000 allow us to focus on the within-country variation in income inequality. Since the main sources of income for the vast majority of residents of post-Soviet countries are labour income and social transfers, the focus of our research is income inequality. Our baseline model examines the impact of economic growth and changes in the labour share on income inequality. We use fixed-effects two-stage least squares for panel data as the primary estimation method, which allows for the endogeneity of the regressors in the central equation (Semykina and Wooldridge 2010), taking into account the results of previous studies showing the two-way relationship between inequality and economic growth.

After this introduction, Section 2 presents a literature review on the relationships among income inequality, labour share and economic growth. Section 3.1 describes the data and Section 3.2 explains the empirical framework applied for econometric modelling. The main findings are in Section 4, while Section 5 briefly discusses the results compared to the theoretical expectations. Section 6 concludes and proposes direction for further research.

## 2. Literature Review

The discussion surrounding inequality typically distinguishes between inequality of outcomes, as measured by income, wealth or expenditure, and inequality of opportunity, attributed to factors that exceed individual control. Inequality of outcomes results from the interaction between opportunities and an individual's efforts, and that is why it is difficult to isolate opportunity from effort (Corak 2013; Autor 2014); therefore, most of the academic research focuses on income inequality or wealth inequality.

The cash flow from wages and salaries is the main source of income for most people, especially in developing economies. Over time, regular income from employment enables people to own assets such as a home or a financial portfolio for retirement. Thus, income inequality can transform into wealth inequality. However, in the academic literature, income inequality is the commonly cited type of inequality. One reason for this is that income inequality is measured using well-developed metrics—the market (gross) and net (after tax and social security transfers) Gini indices

Nevertheless, wealth inequality becomes particularly significant when accumulated assets are used as capital. As T. Piketty argues that capital income tends to be more unequally distributed than labour income, an increase in the capital share would likely lead to increased overall income (and, over time, wealth) inequality (Piketty 2014). For this

reason, most academics, as well as international organisations and national governments, currently pay great attention to monitoring the share of labour in the national income.

In recent years, the issue of the declining labour share in the national income of several countries worldwide since the 1980s has received increasing attention in the academic literature. For example, Koh et al. (Koh et al. 2020) analysed long time-series for the United States, Canada, France, Denmark, Sweden, and Japan. They discovered a significant decrease in the labour share in each of the five countries. Kramer (2011) reported a decline in the labour share among the G-7 countries from a peak of 74 per cent in 1974 to 64 per cent in 2010. Meanwhile, Maarek (2012) found that the labour share of income fell by 10 per cent in relation to the GDP between 1980 and 2000 in developing countries with lower and lower-middle incomes.

Additionally, van Treeck (2020) has argued that, since the early 1990s, labour's relative income has been declining by an average of 11 per cent in 90 low and middle-income countries. Diwan (2001) found that the decline in the labour share in Latin American countries began in 1982. In contrast, in African countries, it began in 1975. The rationales behind the worries of scientists and politicians regarding this phenomenon are lucidly explained in Thomas Piketty's seminal work, *Capital in the Twenty-First Century* (Piketty 2014), which proves that the observed decline in the labour share may engender the growth of inequality due to "the private rate of return on capital being significantly higher for long periods than the rate of growth of income and output" and the wealth accumulated growing more rapidly than output and wages in the past. Besides fuelling social tensions, the increasing inequality could significantly hinder sustainable economic growth (Matyushok and Balashova 2021).

In the past two decades, more empirical evidence has emerged regarding the relationship between the labour share of income and income inequality. For example, research by Francese and Mulas-Granados (2015) showed that the labour share is a decisive factor in wage inequality, drawing from panel data of 93 countries from 1970 to 2013. M. Dao and co-authors (Dao et al. 2017) argued that a reduction in the labour share is associated with an increase in the Gini coefficient, as evidenced by a panel of 49 countries (31 advanced economies and 18 emerging market economies) between 1991 and 2014. Sauer et al. (Sauer et al. 2020) reached the same conclusion using a panel for 73 countries (primarily advanced OECD countries) between 1981 and 2010, and Erauskin (2020) found the relationship between a declining labour share and increasing income inequality in 40 of the 62 examined developed and developing countries in the period 1990–2015. However, the opposite direction was found in 22 of the 62 countries. Examining 16 Latin American countries over the period 1950–2012, Alargo (Alargo 2016) found that redistribution policies targeted at wages are conducive to economic growth in Latin America and reduce inequality.

The recent literature on the decrease in labour shares has concentrated on examining various factors, including globalisation's influence (Decreus and Maarek 2008; Maarek 2012; Brada 2013; Kramarz 2017; van Treeck and Wacker 2020; Tian et al. 2022), development of financial capitalism (Pariboni and Tridico 2019; Alexiou et al. 2022; Khan et al. 2022), high natural resource rents (Brada 2013; Al-Marhubi 2021), and technological and structural change (Briguglio and Vella 2016; Acemoglu and Restrepo 2018; Koh et al. 2020). In the global economy, capital (especially portfolio investments) is becoming more mobile than labour, which lowers the bargaining power of workers in the home country because it takes into account the lower wages that are available in other countries where the owner of the financial or physical capital can move his production (Hummels et al. 2014; Kramarz 2017; O'Mahony et al. 2021; van Treeck and Wacker 2020; Diwan 2001; Braakmann and Brandl 2021). Technological progress also augments capital and decreases the labour share (Briguglio and Vella 2016; Guimarães and Mazeda Gil 2022). In addition, the automation of jobs leads to increased losses of jobs in middle-skilled occupations (Autor and Dorn 2013; Autor et al. 2016). Significant income from natural resources exports can cause a decline in traditional manufacturing sectors, which, in turn, is related to lower labour shares (Al-Marhubi 2021).

Both theoretical and empirical studies also witness that, in addition to the declining labour share, other factors driving inequality could be high inflation (Siami-Namini and Hudson 2019; Liosi and Spyrou 2022; Tomkiewicz 2018), poor governance (Bahamonde and Trasberg 2021), low financial development (Kim 2016) and some structural factors such as the share of the shadow economy (Yap et al. 2018) or share of manufacturing employment compared to agriculture and service (Pariboni and Tridico 2019; Vo et al. 2019). Conversely, effective government can and should reduce inequality in several ways. For example, the Scandinavian approach is known for its generous welfare and progressive tax policies. In contrast, the coordinated market economy approach (mostly EU countries) uses more stringent financial regulation and more capital-intensive strategies in the manufacturing sector (Pariboni and Tridico 2019) in order to curb rising inequality. However, there is still considerable uncertainty concerning the government's efficiency in the fight against inequality. The logical assumption that democratic governments can better deal with rising inequality is not always supported by empirical evidence. Thus, Bahamonde and Trasberg (2021) argue that democratisation and democratic rule in the context of high state infrastructural power are associated with increases in income inequality based on panel regressions for 126 industrial and developing countries for 1970 and 2013. They state that a good democracy attracts investors; investors bring capital and develop the financial sector, contributing to the growth of incomes and, as a result, the growth of inequality.

Regarding studies devoted to investigating the relationship between inequality and economic growth (when the dependent variable is inequality or economic growth), one can note that the interest of scientists in this problem does not decrease over time. The type of relationship between economic growth and inequality, as proposed by S. Kuznets (Kuznets 1955), has been repeatedly tested on various empirical data with different results. Mdingi and Ho (2021) provided a thorough literature review and summarised the results of empirical studies on income inequality and economic growth with positive and negative relations and with no relations. Rubin and Segal (2015) discovered that from 1953 to 2008, income inequality was positively linked to economic growth in the U.S., while Cingano's (2014) findings indicated a negative correlation between income inequality and economic growth in OECD countries. Fawaz et al. (2014) concluded the same for low-income developing countries.

An increasing number of recent studies apply more sophisticated means of analysis in order to find the relationship between inequality and economic growth and include other factors in the consideration, such as initial income (Brueckner and Lederman 2018), financial inclusion and development (Kim 2016; Madsen et al. 2018), public investments (Turnovsky 2015), and innovation and human capital (Adrián Risso and Sánchez Carrera 2019).

Therefore, this literature review shows that, although the relationship between economic growth, labour share and income inequality has received necessary attention recently, the evidence has been far from conclusive. This is especially true for post-Soviet countries because, first, this group of countries has inherited several distinctive features in terms of economic development. Second, a few researchers have addressed the problem of inequality and economic growth in post-Soviet countries. All the countries started the late 1980s with shallow levels of inequality, with the Gini index being, on average, below 25 per cent, which remained below 30 per cent for the early 1990s and then increased dramatically afterwards. Inequality later stabilised at around 33 per cent for countries that were new members of the EU (Baltic countries) and around an average of 40 per cent for the rest of the former USSR republics, with significant heterogeneity across the countries (Grimalda et al. 2010; Habibov 2013; Thorez 2014) and even across the regions in large countries like Russia, Ukraine and Kazakhstan (Zubarevich and Safronov 2011). High income inequality was accompanied in most of the post-Soviet countries by growing inequality in access to health care (Rusinova and Brown 2003), education (Konstantinovskiy 2012; Ibragimova and Frants 2020), childcare system (Kosyakova and Yastrebov 2017), and social inclusion (Spoor 2018).

Another important distinctive feature of most post-Soviet countries is market imperfections, such as a missed connection between the unemployment level and the level of income inequalities (Tomkiewicz 2018). Artificially stifled unemployment is accompanied by significant worsening of the working conditions and low wages. The second feature manifested most of all during the formation of national economies is that an increase in income inequalities does not translate into a growth of productivity of the economy, with an exemption for new EU members (Grimalda et al. 2010). Tomkiewicz argues that increasing income inequalities do not necessarily result from healthy market forces but point towards imperfections. This is the case in some countries, like the Russian Federation, where a small number of economic agents can derive massive benefits from their privileged position while most of the society remains poor. The post-Soviet countries have already gone through about 30 years of development, following various economic and political models, which makes it possible to compare their effectiveness in achieving sustainable development goals and combating inequality. To the best of our knowledge, there are very few examples of quantitative studies in the literature investigating factors that impact inequality in the group of post-Soviet countries (Grimalda et al. 2010; Náplava 2020). In addition, the main limitation of the existing literature is that it focuses on the transitional countries of Eastern and Central Europe.

Therefore, this paper contributes to the literature by providing empirical evidence of the relationship between income inequality, labour share, government efficiency and economic growth in post-Soviet countries from 2000 to 2020.

## 3. Methodology

### 3.1. Data Description

There are 15 post-Soviet states in total: Armenia, Azerbaijan, Belarus, Estonia, Georgia, Kazakhstan, Kyrgyzstan, Latvia, Lithuania, Moldova, Russia, Tajikistan, Turkmenistan, Ukraine, and Uzbekistan. The present study applies an unbalanced panel dataset for 11 post-Soviet countries from 2000 to 2020. We excluded Azerbaijan, Tajikistan, Turkmenistan, and Uzbekistan due to the lack of some data in the international databases. We view the first ten years after the collapse of the Soviet Union in 1991 as years of transition from command economies to other forms of economies. Our objective is to examine the impact of economic growth, government expenditures, and government policies on income inequality during the post-transition era.

Our dataset includes the following variables:

1. Inequality. The Gini index measures income inequality. Data are derived from Solt's Standardized World Income Inequality Database (SWIID), Version 9.5 (Solt 2020). We use two measures: inequality in disposable (post-tax, post-transfer) income, denoted as GINI_DISP, and the share of income held by quintiles. The data on the percentage share of income or consumption that accrues to quintiles subgroups of the population are from the World Development Indicators (WDIs). To check the robustness of our results, we take the GINI index (World Bank estimate) from the World Development Indicators (denoted as GINI_WB).

2. Economic growth. The variable GDP measures the per capita GDP (PCGDP) per capita PPP (constant 2017 international dollars) provided by the WDI database. Data on GDP growth (denoted as GDP, measured as changes in the natural log) are retrieved from the Total Economy Database™ (TED), version April 2023. The TED is a database with annual data concerning the GDP, population, employment, hours, labour quality, capital services, labour productivity, and total factor productivity for 131 countries.

3. Labour share. The TED labour share (LSH) is calculated as the share of compensation of workers (including the self-employed) in relation to the nominal GDP at market prices.

4. As instrumental variables for economic growth, we use the Labour input and Capital input obtained from the TED. We use data for the Growth of Labour Quantity change in the natural log (LINQNT) and Growth of Labour Quality, the change in the

natural log (LINQLT). In different exercises, we use the Growth of Total Capital Services, change in the natural log (denoted as TOTCAP), or Growth of Capital Services provided by Non-ICT Assets, the change in the natural log (denoted as NONICTCAP).

5.　We consider some potential variables mediating the relationship between income and inequality. The government expenditure (XGOVEXP) is the general government's final consumption expenditure (per cent of GDP) from the World Development Indicators. We also consider the government expenditure on education (per cent of GDP) from the WDIs.

6.　We incorporate two dummy variables (D_CD and D_CA), considering the state of democracy evaluated by Freedom House. Belarus, Kazakhstan, Kyrgyzstan, and Russia are considered Consolidated Authoritarian Regimes (D_CA = 1 for these countries and 0 for the rest). The Baltic states (Estonia, Latvia, and Lithuania) are Consolidated Democracy States (consequently, D_CD = 1 for these countries and 0 for the rest). The rest of the countries are evaluated as SAC (with a Semi-Consolidated Authoritarian Regime) or T/H (with a Transitional/Hybrid Regime).

7.　We include a dummy variable (HMIC) considering the World Bank classification of countries by their income. Low Income and Low-Middle Income take a value of zero. Upper-Middle Income and High Income take a value of one. We take into account that some of the countries in question moved into the higher-income group, or vice versa, over the observation period.

Below, we describe our primary variables of interest.

### 3.1.1. Income Inequality

In nearly all the countries analysed, inequality levels have decreased over the past two decades. Taxes and transfers considerably reduce the level of inequality, as evidenced by calculations based on pre-tax data. Our research employs the variable GINI_DISP as it is most fitting to assess the efforts of public policy in reducing inequality. The data from the World Bank, while not always available and more prone to fluctuations, align with the GINI_DISP for specific countries (see Figure 1).

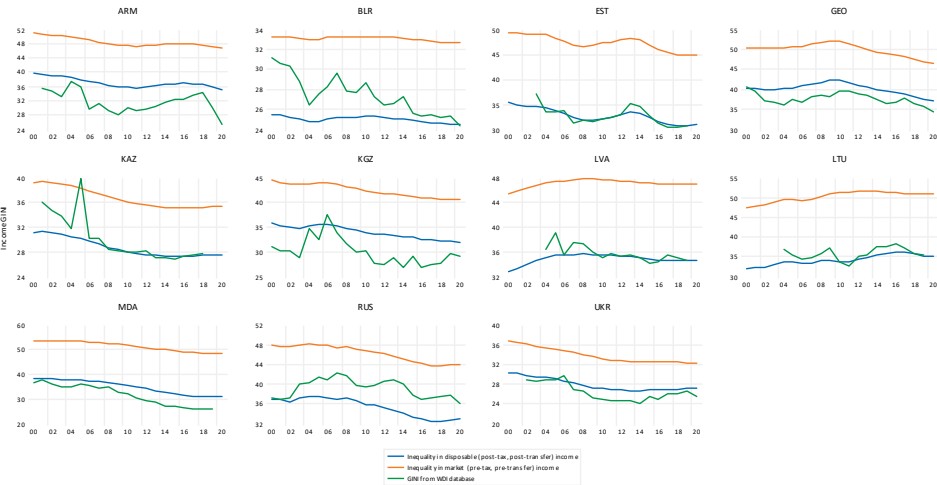

**Figure 1.** Different measures of income inequality[1]. Source: SWIID, WDI database.

The analysed countries display heterogeneity regarding inequality. Notably, inequality rates in Belarus are relatively low, remaining stable with slight changes in the Gini coefficient throughout the study interval. By contrast, Moldova's Gini coefficient demonstrates considerable variation during the same time frame (see Figure 2).

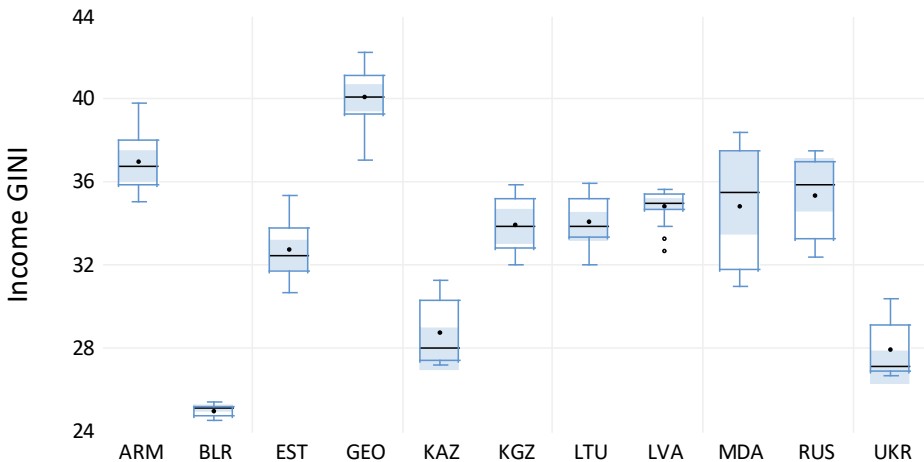

**Figure 2.** Boxplot of the GINI coefficient (inequality in disposable (post-tax, post-transfer) income). Source: Authors' calculation based on SWIID. Here black dot is mean, black line is median, white dot is near outliers, shade- 95% confidence for median.

As indicated in Figure 2, on average, the most significant inequality is observed in Georgia, followed by Armenia, Russia, and Moldova.

3.1.2. Labour Share

The proportion of labour in the GDP was relatively elevated in the Soviet Union. However, it significantly declined in the early 1990s across most nations within the ex-USSR region. Despite this, it remained higher, on average, than in the early 2000s. As of 2020, a labour share below 50 per cent (LSH < 50) was recorded in Armenia, Georgia, Kazakhstan, and Russia (Figure 3).

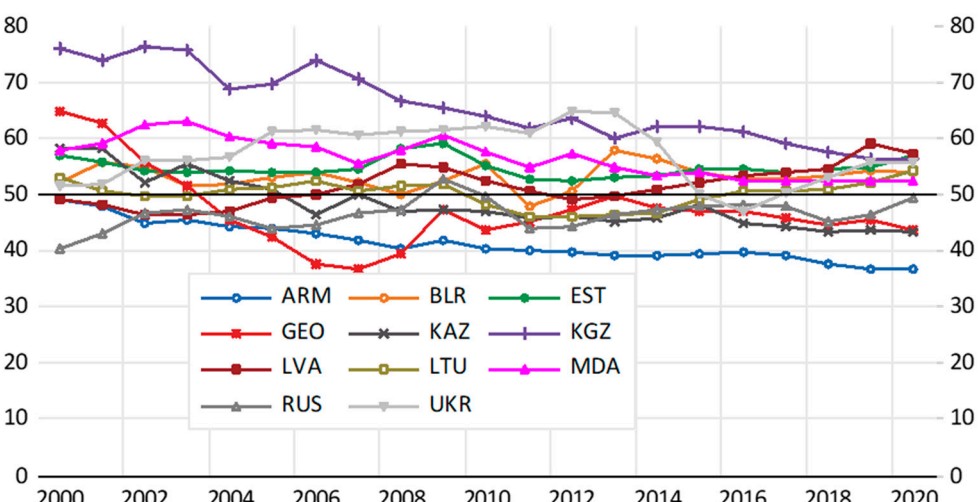

**Figure 3.** Dynamics of labour share of GDP in post-Soviet countries in 2000–2020. Source: TED.

The labour share of income is positively associated with the Gini coefficient, on average (Figure 4).

It is not implied that there is a causative link between an increase in the labour share and an increase in inequality. However, there is a positive correlation. Looking at the income shares of the first to fifth quintiles, it is apparent that an increase in the labour share is correlated with an increase in the income share of the wealthiest individuals and a decrease in the share of the first and second quintiles (see Figure 5).

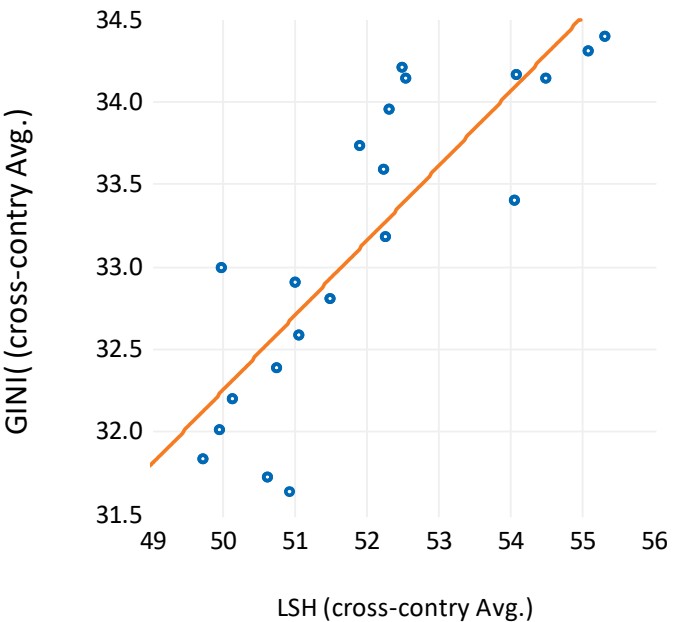

**Figure 4.** Scatter diagram between the Gini coefficient and labour share of income (cross-country average). Source: Authors' calculations based on SWIID and TED.

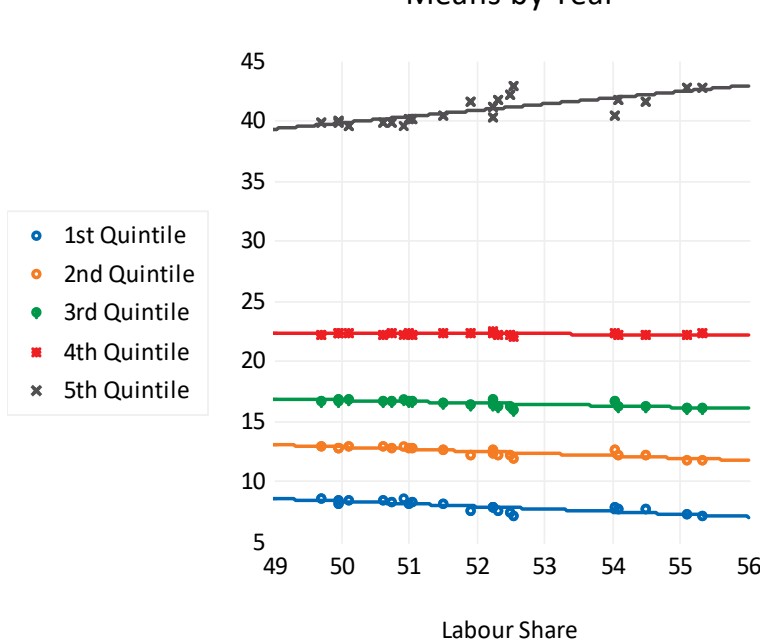

**Figure 5.** Scatter diagram displays the cross-country average for each year from 2000 to 2020 of the labour share vs. income quintiles. Source: Authors' calculations based on WDI and TED.

3.1.3. GDP per Capita

The GDP per capita steadily grew in all the countries, except during the global financial and economic crisis, but at varying rates (Figure 6). The pace of economic growth in the aftermath of the crisis has also been uneven.

Figure 7 presents the simple correlation between national income and inequality. The relationship is negative between the cross-countries averages of the GINI and GDP per capita. The graph illustrates that, as income rises, the share of the upper quintile decreases while that of the bottom quintile increases.

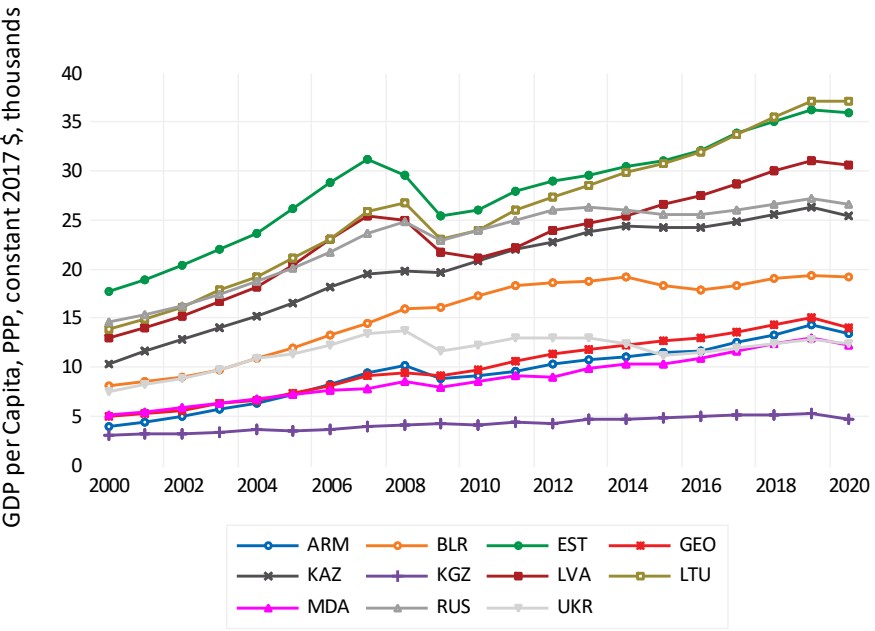

**Figure 6.** Income growth. Source: WDI database.

However, this is just a simple correlation between the averages and a regression analysis is necessary.

### 3.2. Empirical Framework

Our baseline regression relates income inequality to the labour share of income and real GDP per capita:

$$\text{Inequality}_{it} = \alpha_i + \beta \text{Log(LSH)}_{it} + \gamma \text{Log(PCGDP)}_{it} + u_{it}, \qquad (1)$$

where $\alpha_i$ are country fixed effects that control for cross-country time-invariant determinants of income and its distribution. Here, $t$ equals one for 2000 and $u_{it}$ is the error, which varies over $i$ and $t$. Following the empirical growth literature, we use the real GDP per capita in levels as in (Dollar and Kraay 2002; Brueckner et al. 2014) and estimate Equation (1) using panel data. The sample period is 2000–2020 and covers 11 countries. Table A1 presents the descriptive statistics of our variables, while Table A2 contains the list of countries in our sample.

This model explains what happens to income distribution as the log of national income changes and/or the share of national income allocated to labour compensation changes. We employ logarithmic transformation of the data in order to create a smoothed data series and assume a non-linear relationship between the dependent and independent variables. To test the validity of this assumption, we use the Box–Cox test to determine whether the model should utilise non-transformed variables or the transformed model (1).

Although using the panel data has the apparent benefit of increasing the number of observations, it may violate at least two fundamental assumptions underlying ordinary least squares (OLS) estimation. The temporal structure of the data increases the chance of autocorrelation, and the cross-sectional design of the data increases the possibility that the variance in the error terms may differ across countries and cause a heteroscedasticity problem. The consequence of these violations is that OLS coefficient estimates are still unbiased but inefficient. To rectify these issues, we have adopted the Panel-Corrected Standard Errors (PSCE) method as per the recommendation of Beck and Katz (1995) and detailed in Adrián Risso and Sánchez Carrera (2019).

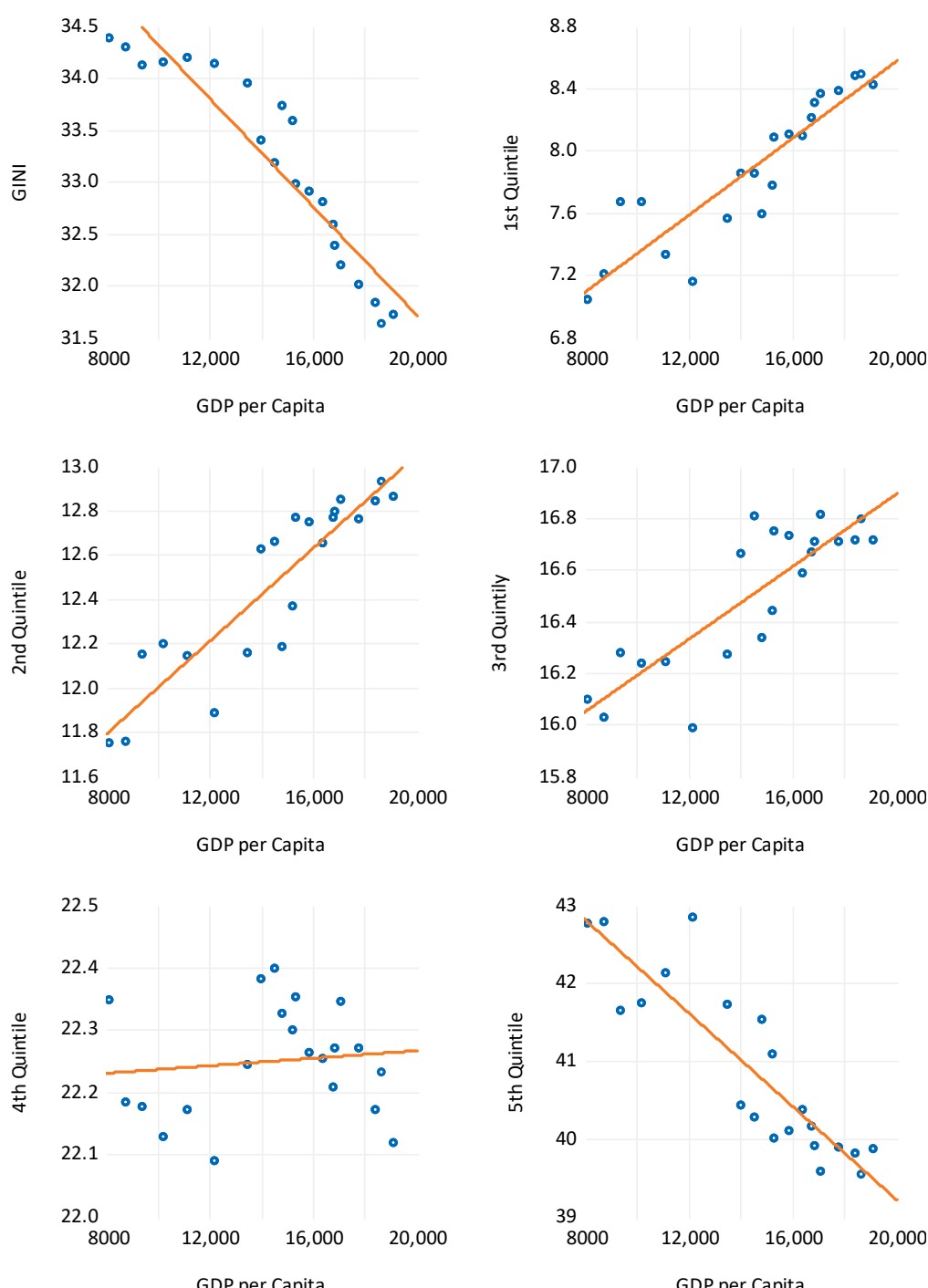

**Figure 7.** Inequality and GDP per capita. Source: Authors' calculation based on WDI database.

As was indicated in (Brueckner et al. 2014), countries' GDP per capita is endogenous in model (1) because inequality can affect economic growth (Brueckner and Lederman 2018). Our study did not find Granger causality between inequality and the GDP per capita. However, there may be a correlation between the GDP per capita and the error term $u_{it}$ in Equation (1), leading to endogeneity bias.

There is a possibility that the labour share of income is also endogenous in Equation (1). To address this issue, we apply the Hausman test for the panel data (Wooldridge 2010). We first regress $Log(LSH)_{it}$ on $Log(PCGDP)_{it}$ using fixed-effect OLS. Then, we re-estimate Equation (1) using the residuals from this regression instead of the variable $Log(LSH)_{it}$ and test the corresponding coefficient for significance. The same is performed for the variable

Log(PCGDP)$_{it}$. The results show that we can regard Log(PCGDP)$_{it}$ as endogenous and Log(LSH)$_{it}$ as exogeneous in Equation (1).

We employ the two-stage least squares (TSLS) estimation for endogeneity biases. The corresponding first-stage equation is as follows:

$$\text{Log(PCGDP)}_{it} = a_i + b \cdot \text{NONICTCAP}_{it} + c \cdot \text{LIQNT}_{it} + e_{it}, \tag{2}$$

The assumption in TSLS is that the growth of labour quantity (LIQNT) and growth of capital services provided by non-ICT assets (NONICTCAP) affect income inequality through their effect on the GDP per capita. To assess the instruments' validity, the Hansen J test is employed.

In our research, our assumption is that government spending may impact inequality in different ways depending on a country's wealth and political structure. An individual's income tends to increase with a higher level of education (Rupert et al. 1996; Murphy and Welch 1990). Our model considers government spending on education and efficiency as factors contributing to reducing inequality.

The model can be written as follows:

$$\text{Inequality}_{it} = \alpha_i + \beta \text{Log(LSH)}_{it} + \gamma \text{Log(PCGDP)}_{it} + \sum \delta_j X_j + u_{it}, \tag{3}$$

where $X_j$ are other factors that are assumed to impact inequality: total government expenditure, government expenditure on education, the government effectiveness, the democracy level, and income level.

To address the potential problem of heteroscedasticity and serial autocorrelation in panel data, we use Panel-Corrected Standard Error (PCSE) methodology (Beck and Katz 1995). The cross-section weights (PCSE) and period weights (PCSE) are robust to heteroskedasticity across cross-sections or periods, respectively.

## 4. Regression Results

### 4.1. Baseline Model

As a preliminary step, we examine the independent variables in our baseline model for endogeneity. The results indicate that the labour share of income is exogenous. However, the GDP per capita is endogenous. Therefore, we use instruments to address the issue of endogeneity. However, we provide OLS and TSLS estimation results (Table 1, Panel A and B consequently) for a robustness check and to determine compatibility with other models.

The main result of the TSLS estimation (Table 1, Panel B) is that exogenous within-country variations in the GDP per capita are negatively related to within-country variations in income inequality.

For our baseline model, the results show that, on average, a one per cent increase in the GDP per capita reduces the Gini coefficient by 0.11 per cent when controlling for the labour share and country fixed effect (Table 1, Panel B, col. 1). For example, for such a country as Russia with a (sample average) Gini coefficient of 35.4 and a (sample average) PPP GDP per capita of $22,870, one would expect the Gini coefficient to decline to 35.0 as the income per capita increases by 10 per cent. The effect is not very large but significant. Note that the within-country increase in the GDP per capita, controlling for the labour share, has a positive and significant impact on the income share in the low income group (1st and 2nd quintiles) but a negative impact for the 4th and 5th quintiles (Table 1, col. 2–6).

As for the effect of the labour share on inequality, controlling for national income and country fixed effects, it is estimated to be positive and insignificant for the Gini (panel B, Table 1). If we apply LS to our model, the impact of the labour share is positive and significant (Panel A, Table 1).

These findings challenge the established theory (which suggests that an increase in the share of capital and a decrease in the percentage of labour should lead to a rise in inequality, given that there are far fewer capital owners than workers) and contradicts some empirical

evidence. Nevertheless, (Erauskin 2020) reports that, in certain countries, an increase in the labour share has coincided with an increase in inequality.

**Table 1.** Effects of labour share and national income on national income inequality.

| | Gini | 1st Quintile | 2nd Quintile | 3rd Quintile | 4th Quintile | 5th Quintile |
|---|---|---|---|---|---|---|
| | (1) | (2) | (3) | (4) | (5) | (6) |
| | | | Panel A: LS (PCSE) | | | |
| Log(LSH) | 0.09 ** | −0.89 * | −1.57 *** | −1.38 *** | −0.70 ** | 4.46 *** |
| | (0.03) | (0.51) | (0.51) | (0.42) | (0.34) | (1.53) |
| Log(PCGDP) | −0.08 *** | 1.91 *** | 1.19 *** | 0.73 *** | −0.24 * | −3.55 *** |
| | (0.01) | (0.18) | (0.18) | (0.15) | (0.13) | (0.55) |
| R-squared | 0.93 | 0.85 | 0.80 | 0.76 | 0.47 | 0.79 |
| | | | Panel B: TSLS (PCSE) | | | |
| Log(LSH) | 0.04 | −0.89 | −1.59 ** | −1.73 *** | −1.57 *** | 5.34 ** |
| | (0.04) | (0.75) | (0.70) | (0.61) | (0.61) | (2.14) |
| Log(PCGDP) | −0.11 *** | 1.92 *** | 1.18 ** | 0.43 | −0.94 ** | −2.86 ** |
| | (0.03) | (0.51) | (0.45) | (0.37) | (0.37) | (1.34) |
| Hansen J, *p*-value | 0.71 | 0.98 | 0.43 | 0.18 | 0.40 | 0.40 |
| First-stage Fstat | 23.2 | 17.6 | 23.2 | 23.2 | 23.2 | 23.2 |
| Country FE | Yes | Yes | Yes | Yes | Yes | Yes |
| Observations | 231 | 213 | 213 | 213 | 213 | 213 |

Notes. The dependent variable is Log(GINI_DISP). The estimation method in Panel A is least squares; Panel B is two-stage least squares. Cross-section weights (PCSE) standard errors and covariance (d.f. corrected) are shown in parenthesis. The instruments in Panel B are NONICTCAP and LIQNT, except for the regression for the 1st quintile group where total capital service growth is used instead of NONICTCAP. * Significantly different from zero at the ten per cent significance level, ** five per cent significance level, *** one per cent significance level.

However, the labour share effect is multi-directional and significant for the quintile groups. Controlling for national income, a labour share increase is associated with a decrease in the income share of low- and middle-income groups. Meanwhile, the percentage of the wealthiest group increases with an increase in the labour share.

It is well known that OLS is typically inconsistent in the presence of an endogenous regressor. Therefore, TSLS is generally preferred in its place. However, in the case of our baseline model, OLS and TSLS produce comparable results. We adhere to established practice and present the Hansen J test to assess the instruments' validity. The null hypothesis is that the instruments are uncorrelated with the error term in the second stage. The p-value of the Hansen J test exceeds 0.1, indicating that we cannot reject the hypothesis that the instruments are valid. We calculate the first-stage F-statistic, with the rule of thumb being that a value less than 10 indicates a weak set of instruments and biased TSLS estimation. Table 1 displays that all the regressions have a first-stage F-statistic > 10.

To address the potential autocorrelation in the residuals, we estimate model (1) using the period weights PCSE. The findings closely resemble those in Table 1 and do not alter our main conclusions.

### 4.2. Model with Government Impact on Inequality

The impacts of government policies on inequality are widely debated. While some argue that progressive taxation and social welfare programmes can help to reduce inequality, others believe that these policies can discourage economic growth and ultimately harm the disadvantaged. Nevertheless, empirical evidence suggests that the impact of government policies on inequality is complex and varies across countries and periods. Therefore, a comprehensive analysis of specific policies in specific contexts is necessary to fully understand their impact on inequality.

In Table 2, we present estimates of the econometric model specified in Equation (3), which includes the general government expenditure (% GDP) (col. 1), an interaction between government expenditure and the type of political system (col. 2), an interaction of

government expenditure and income group (col. 3), and an interaction of the government expenditure on education (% GDP) and a type of political system (col. 4).

**Table 2.** Effects of government expenditure on national income inequality.

| | Government Expenditure | Government Expenditure and Type of Political System | Government Expenditure and Income Group | Government Expenditure on Education and Type of Political System |
|---|---|---|---|---|
| | **(1)** | **(2)** | **(3)** | **(4)** |
| | | Panel A: LS (PCSE) | | |
| Log(LSH) | 0.09 ** (0.04) | 0.13 *** (0.03) | 0.09 ** (0.04) | 0.15 *** (0.04) |
| Log(PCGDP) | −0.08 *** (0.01) | −0.11 *** (0.01) | −0.08 *** (0.01) | −0.09 *** (0.01) |
| Log(XGOVEXP) | 0.004 (0.02) | 0.15 *** (0.025) | 0.005 (0.003) | --- |
| Log(XGOVEXP)*DCA | --- | −0.27 *** (0.04) | --- | --- |
| Log(XGOVEXP)*DCD | --- | −0.63 *** (0.06) | --- | --- |
| Log(XGOVEXP)*HMIC | --- | --- | −0.002 (0.002) | --- |
| XGOVEDUC | | | | 0.014 ** (0.006) |
| XGOVEDUC*DCA | | | | −0.015 *** (0.009) |
| XGOVEDUC*DCD | | | | −0.05 *** (0.01) |
| Country FE | yes | Yes | Yes | yes |
| Observations | 231 | 231 | 231 | 231 |
| R-squared | 0.93 | 0.95 | 0.93 | 0.94 |
| | | Panel B: TSLS (PCSE) | | |
| Log(LSH) | 0.03 (0.05) | 0.07 * (0.04) | 0.03 (0.05) | −0.06 (0.06) |
| Log(PCGDP) | −0.13 *** (0.03) | −0.18 *** (0.02) | −0.14 *** (0.04) | −0.23 *** (0.04) |
| Log(XGOVEXP) | −0.005 (0.02) | 0.19 *** (0.03) | 0.002 (0.02) | |
| Log(XGOVEXP)*DCA | --- | −0.38 *** (0.05) | --- | |
| Log(XGOVEXP)*DCD | --- | −0.78 *** (0.09) | --- | --- |
| Log(XGOVEXP)*HMIC | --- | --- | 0.01 (0.007) | --- |
| XGOVEDUC | --- | --- | | 0.02 (0.006) |
| XGOVEDUC*DCA | --- | --- | | −0.01 * (0.007) |
| XGOVEDUC*DCD | --- | --- | | −0.08 *** (0.02) |
| Country FE | yes | Yes | yes | Yes |
| Observations | | 231 | | 210 |
| Hansen J, p-value | 0.75 | 0.41 | 0.26 | 0.78 |
| First-stage Fstat | 22.2 | 29.5 | 15.6 | 14.5 |

Notes: The dependent variable is Log(GINI_DISP). The estimation method in Panel A is least squares; Panel B is two-stage least squares. Cross-section weights (PCSE) standard errors and covariance (d.f. corrected) are shown in parenthesis. The instruments in Panel B are Growth of Capital Services provided by Non-ICT Assets and Growth of Labour Quality, except for the regression for the 2nd and 4th models where the GDP per capita of the previous period and growth of total capital service are used. * Significantly different from zero at the ten per cent significance level, ** five per cent significance level, *** one per cent significance level.

The findings reveal that, assuming this effect is universal for the studied nations, general government expenditure does not considerably impact inequality. Current levels of national income do not modify these results. The empirical evidence does not support the hypothesis that increased public spending reduces inequality more for middle- and high-income countries than for low-income countries (as classified by the World Bank).

Government spending has varying effects on inequality based on the political system in place, at least according to Freedom House's definition of political systems. Developed democracies such as Estonia, Latvia, and Lithuania demonstrate a negative correlation between public spending and inequality. For instance, based on our calculations (column 2 of Table 2, Panel B), if we consider a nation such as Estonia that has an average Gini coefficient of 32.64, an average PPP GDP per capita of $28,130, a labour share of the GDP of 54.76, and government expenditure of 18.6 per cent of GDP, we anticipate a 0.5 per cent decrease in the Gini coefficient as the government expenditure (% of GDP) rises by 1 per cent. We expect to see the same outcome in other Baltic states. For nations where power is predominantly concentrated in the hands of the leader (Russia, Belarus, Kazakhstan, Kyrgyzstan), estimates suggest a decrease in income inequality as general government expenditure increases, albeit to a lesser degree. Conversely, in countries with mixed political regimes (Armenia, Georgia, Moldova, Ukraine), the results indicate that an increase in government expenditure (while accounting for the per capita GDP and labour income share) corresponds with a rise in income inequality.

We can assume that EU membership is a latent variable that influences the impact of government spending on inequality. In our sample, only EU members possess developed democratic institutions. Therefore, the effects of democratic institutions and EU membership cannot be distinguished in our study.

The literature indicates that nations with more developed educational systems exhibit lower levels of economic inequality (Abdullah et al. 2015; Coady and Dizioli 2018; Reinders et al. 2021). Thus, we investigate the hypothesis that public spending on education has an impact on the degree of inequality.

Again, the empirical findings indicate that increased government spending on education is expected to decrease inequality only in the Baltic nations and, to a lesser extent, in Russia, Belarus, Kazakhstan, and Kyrgyzstan. There is no evidence to suggest that government expenditure on education tends to reduce inequality in the other countries under consideration (refer to Table 2, column 4).

By itself, the level of government expenditure does not guarantee the attainment of a government's objectives, irrespective of the political system in place. The World Bank assesses governmental effectiveness using a distinct methodology (Kaufmann et al. 2011). More specifically, government effectiveness is an aggregate indicator combining the views of many enterprise, citizen and expert survey respondents from diverse countries and ranges from −2.5 to 2.5.

We utilise this indicator as outlined in (Náplava 2020) to objectively assess the effectiveness of the government in reducing inequality. We evaluate an equation incorporating government efficiency (controlling for the labour share and per capita income). In these exercises, the dependent variable is the Gini coefficient from the World Bank and he income quintile shares sourced from the same institution. The estimation results are presented in Table 3.

The findings indicate that augmented government efficacy would lead to a decline in inequality.

For instance, our calculations (col. 1 of Table 3) reveal that in a country like Estonia, possessing an average Gini coefficient of 32.64, PPP GDP per capita of $28,130, and government effectiveness index of 1.022, there should be a decrease of 0.2 points in the Gini coefficient when the government effectiveness increases by 0.1 points (the related regression coefficient being −2.03 and significant at a 1% level of significance).

As government efficiency improves, it is estimated that the share of national income going to the poorest segments of the population will increase while the share going to the richest will decrease.

Again, we anticipate a rise in inequality as the labour share increases, all other variables being constant. This has a counterintuitive impact. However, examining how the labour share influences the income shares of the quintile groups, we discover that an increase in the labour share, without growth in the GDP per capita or government efficiency, is

liable to reduce the share of the first two quintile groups and amplify the wealthiest share. However, examining how the labour share influences the income shares of the quintile groups, we discover that an increase in the labour share, without growth in the GDP per capita or government efficiency, is liable to reduce the shares of the first two quintile groups and amplify the wealthiest share. The share of labour in the GDP can increase through the growth of high-paid workers' and employees' wages (such as top managers of companies or high-ranking civil servants) rather than through labour productivity growth or the attraction of more qualified personnel. Under these circumstances, inequality rises due to poor government performance and stagnant per capita income.

**Table 3.** Effects of government effectiveness on national income inequality.

| | Gini | 1st Quintile | 2nd Quintile | 3rd Quintile | 4th Quintile | 5th Quintile |
|---|---|---|---|---|---|---|
| | (1) | (2) | (3) | (4) | (5) | (6) |
| LSH | 0.15 *** | −0.02 ** | −0.03 *** | −0.002 *** | −0.006 | 0.08 *** |
| | (0.04) | (0.01) | (0.01) | (0.008) | (0.006) | (0.03) |
| Log(PCGDP) | −5.24 *** | 1.01 *** | 0.60 * | 0.43 | −0.07 | −2.03 ** |
| | (0.85) | (0.34) | (0.32) | (0.26) | (0.24) | (0.99) |
| GOVEFF | −2.03 *** | 1.48 *** | 0.95 ** | 0.50 | −0.20 | −2.63 ** |
| | (0.73) | (0.43) | (0.41) | (0.34) | (0.32) | (1.26) |
| Country FE | YES | YES | YES | YES | YES | YES |
| R-squared | 0.82 | 0.86 | 0.81 | 0.77 | 0.48 | 0.80 |
| Obs | 183 | 206 | 206 | 206 | 206 | 206 |

Note: The dependent variable in the first regression is GINI_WB. The method of estimation is least squares with PCSE (shown in parenthesis). * Significantly different from zero at the ten per cent significance level, ** five per cent significance level, *** one per cent significance level.

Figure 8 presents a scatter diagram that depicts the relationship between productivity and the labour share. The relationship between the labour share and productivity is not straightforward and varies across countries. On the one hand, for countries such as Kyrgyzstan, an extremely high labour share can be explained by the large share of the agricultural sector in an economy (more than 14 per cent in 2021) with low productivity. A decrease in the labour share accompanies an increase in productivity. On the other hand, in countries such as Latvia, the growth in the labour share has been in line with more robust productivity growth.

If we estimate the regression between productivity and the labour share, dividing our sample into two groups: with productivity above 15,000 constant 2017 PPP$ per person employed and below or equal to 15,000, we obtain the following results. For low productivity (<15,000), the regression is negative, but for higher productivity, the regression is insignificant. The Chow test supports this finding.

It is worth repeating that increasing the labour share without increasing the GDP per capita does not reduce inequality.

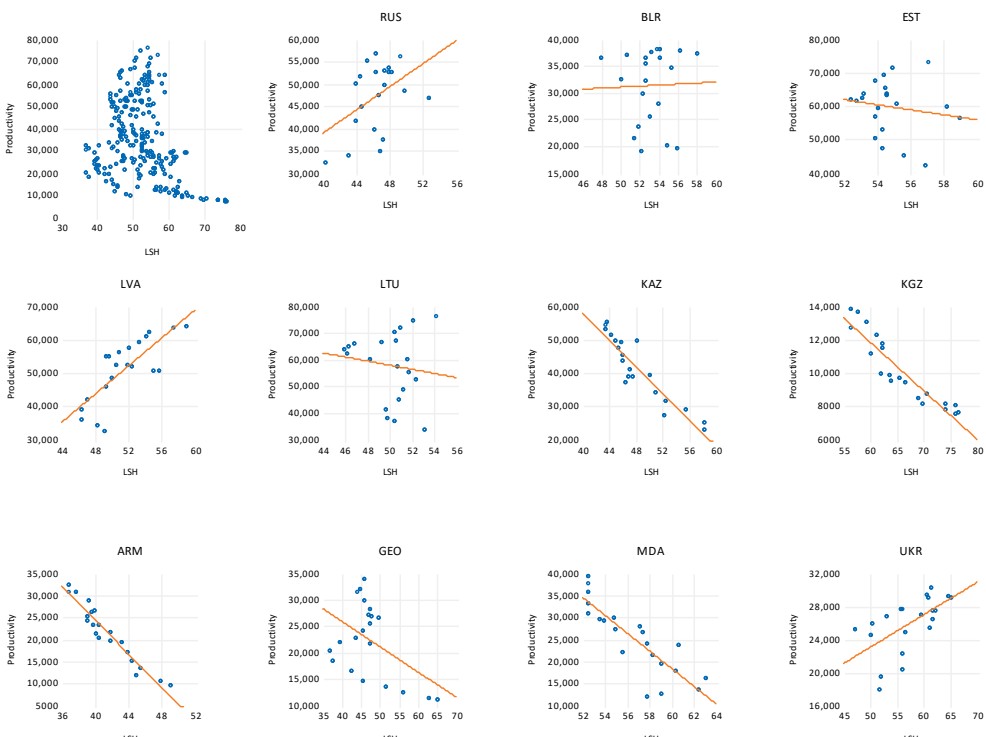

**Figure 8.** Scatter diagram of productivity vs. labour share. Source: Authors' calculations based on data from TED.

### 4.3. Alternative Specifications

We consider several alternative specifications and explore the hypothesis that the labour share is non-linearly related to inequality.

Table 4 presents the results of testing the quadratic relationship between inequality and the labour share for different groups of post-Soviet countries.

**Table 4.** Quadratic dependence of inequality on labour share.

|  | Baseline Model, 11 Countries | Baseline Model, the Baltic States Excluded | Effect of Government Expenditure, 11 Countries | Effect of Government Expenditure, the Baltic States | Effect of Government Expenditure, the Baltic States Excluded |
|---|---|---|---|---|---|
|  | (1) | (2) | (3) | (4) | (5) |
| Log(LSH) | −1.85 *** | −2.09 *** | −1.88 *** | −5.32 | −1.72 *** |
|  | (0.69) | (0.7) | (0.72) | (6.22) | (0.67) |
| Log(LSH)^2 | 0.25 *** | 0.27 *** | 0.25 *** | 0.69 | 0.22 *** |
|  | (0.09) | (0.09) | (0.09) | (0.79) | (0.08) |
| Log(PCGDP) | −0.08 *** | −0.12 *** | −0.08 *** | −0.04 * | −0.12 *** |
|  | (0.01) | (0.01) | (0.01) | (0.02) | (0.01) |
| Log(XGOVEXP) | --- |  | −0.004 | −0.37 *** | 0.05 ** |
|  |  |  | (0.02) | (0.07) | (0.02) |
| Country FE | Yes | Yes | Yes | Yes | Yes |
| R-squared | 0.93 | 0.96 | 0.93 | 0.64 | 0.96 |
| Observations | 231 | 168 | 231 | 63 | 168 |

Note. The dependent variable is Log(Gini_DISP). The method of estimation is least squares with PCSE (shown in parenthesis). * Significantly different from zero at the ten per cent significance level, ** five per cent significance level, *** one per cent significance level.

In the baseline model, a U-shaped curve is observed, i.e., as the labour share increases, inequality decreases and then increases. This is observed for all the countries except the Baltic countries, where no dependence on the labour share was found.

In our sample of countries, the change in the labour share ranges from LSH = 36.6 (Armenia) to LSH = 76.7 (Kyrgyz Republic). According to estimate (5), the top of the parabola corresponds to LSH = 48. Ceteris paribus, we expect inequality to increase in countries with LSH > 48 as the labour share of income rises. It is worth noting that, by 2020, only three countries, Armenia, Kazakhstan and Georgia, have a labour share below the threshold (Figure 3). Therefore, according to our estimates, reducing the labour share to a threshold of 48 per cent of GDP in the remaining countries will not increase inequality. Consequently, the rising labour share is expected to accompany rising inequality.

Evaluating the impact of government spending separately, controlling for other factors, in the Baltic countries and the other eight countries of the former Soviet Socialist Republic, we confirm the result presented in the previous section. In countries with developed democratic institutions, increased public spending is associated with decreased inequality. In contrast, in other countries, it is related to an increase (coefficient equals −0.37, significant at the 1 per cent level and 0.05 at the 5 per cent level).

## 5. Discussion

Therefore, our main counterintuitive result from the panel regression is that we expect an increase in inequality as the labour share increases after the threshold, holding other variables constant. This empirical result confirms that, in post-Soviet countries, the share of labour in the income received by low- and medium-wage workers decreases, and the share received by workers at the top of the wage distribution increases. This phenomenon is exciting but is not unique and was described in some studies in the U.S. (Bivens and Shierholz 2018), Malaysia (Vo et al. 2019; Ng et al. 2019), China, Canada, Mexico and some others (Erauskin 2020). The possible explanation behind this trend could be, on the one hand, high market concentration and "monopsony power" of employers or, on the other hand, a growing share of traditional low-skilled sectors of the economy (some services or primitive obsolete manufacturing). The highly concentrated structure of the labour market is typical for Russia outside of the major cities (Bignebat 2006; Zhuravleva 2021), Kazakhstan (Rama and Scott 1999; Nurzhan 2015), and other countries of the Eurasian Economic Union (Mityushina et al. 2017).

Our results concerning public spending also find support in the literature. Anderson et al. (2017) review 84 papers and examine the relationship between government spending and income inequality. The study concludes that "in comparison with the Gini coefficient, the relationship tends to be stronger (more negative) when focusing on the share of the richest 10 per cent or 20 per cent in national income, and weaker (less negative) when focusing on the share of the poorest 20 per cent or 40 per cent". This finding has significant implications for policymaking as it indicates that the redistributive effects of government spending have been limited to the upper half of the income distribution, particularly benefiting middle-income groups. The study suggests that the lower half of the distribution has not received the same level of benefits. The definitive answer to the question of whether government spending reduces inequality depends very much on the type of spending and the measure of inequality.

Our results show that improving public institutions' efficiency helps reduce inequality. Not only is the Gini coefficient sensitive to changes in the efficiency indicator but also the income shares of the poorest and the richest. Increasing the efficiency of the state contributes to an increase in the income share of the poorest and, to a more considerable extent, a decrease in the income shares of the richest. Still, it has virtually no effect on the income share of the middle classes.

In the Baltic countries with more developed democratic institutions, which the World Bank considers to be effective, the impact of government spending on reducing inequality is much higher than in countries with a strong central government and less developed democracy. However, in countries with transitional political regimes, public spending does not systematically affect the level of inequality.

The same conclusion holds for public spending on education. Therefore, we can reckon that growth in the labour share of income and growth in government spending, including spending on education, are insufficient to reduce income inequality. On the one hand, these changes must be accompanied by an increase in the GDP per capita and productivity growth and, on the other hand, by a high quality of state institutions.

## 6. Conclusions

The evidence from this study points towards the idea that the growth of the labour share by itself does not guarantee a decrease in inequality in society. The monopsony of the labour market can lead to an increasing gap between low- and medium-skilled and high-skilled workers, even in a growing labour share of income. We hope this result will help solve the difficulties of labour market regulation in post-Soviet countries with inherited geographical and sectorial disproportions and outdated employment patterns.

We also have obtained comprehensive results demonstrating that governments in most post-Soviet countries play a crucial role in controlling inequality. However, there is a significant difference in approaches and efficiencies among different political regimes. Therefore, our research suggests that the policymakers in countries with a strong central governments and transitional political regimes should pay more attention to the structure and the efficiency of government spending, including spending on education.

As the primary principles of policy development, we draw attention to two conclusions resulting from our research. Firstly, the expansion of public institutions improves the efficiency of government spending in terms of addressing issues relating to income inequality. Secondly, in contrast, the bolstering of state capitalism in post-Soviet nations has been ineffective in establishing a "welfare state" or an efficient system for redistributing income within the economy. Post-Soviet countries with state capitalist economies have failed to tackle problems such as monopsony in their labour markets, weakening of workers' bargaining positions, regression, and marginalisation of labour. These results have implications for policymakers not just in these regions but globally, given the increasing desire to expand state influence over economies. The expansion of the state's economic role necessitates increased participation from civil society in supervising and controlling public expenditure.

Our work has some limitations. The most important limitation lies in the data availability and measurements. Although we have tested our results using different specifications and econometric methods, we must admit that finding the ideal instrumental variables for TSLS estimation is a challenging task. The use of other instruments may lead to adjustments in the resulting estimates. Despite this, we believe our work could be a starting point for future investigations of other factors concerning inequality and their relationships with economic growth in post-Soviet countries. In this context, it should be noted that future work by the authors will focus on analysing the impact of natural rent and technological factors on the labour share and inequality.

**Author Contributions:** Conceptualization, B.S.S., S.B. and S.R.; Methodology, S.B.; soft ware, S.B.; validation, S.B.; formal analysis, S.R.; investigation, B.S.S.; resourses, S.R.; data curation, S.B.; writing–original draft preproject administration, paration, B.S.S., S.B. and S.R.; writing—review and editing, B.S.S., S.B. and S.R.; visualization, S.B.; supervision, B.S.S.; project administration, B.S.S. All authors have read and agreed to the published version of the manuscript.

**Funding:** This research received no external funding.

**Institutional Review Board Statement:** Not applicable.

**Informed Consent Statement:** Not applicable.

**Data Availability Statement:** The data presented in this study are available on request from the corresponding author.

**Conflicts of Interest:** The authors declare no conflict of interest.

## Appendix A

**Table A1.** Descriptive statistics of the main variables.

|  | **GINI_DISP** | **PCGDP** | **LSH** | **XGOV_EXP** |
|---|---|---|---|---|
| Mean | 33.16 | 16,295.68 | 52.01 | 0.16 |
| Median | 33.90 | 13,990.84 | 52.02 | 0.17 |
| Maximum | 42.30 | 37,184.45 | 76.28 | 0.24 |
| Minimum | 24.50 | 3078.91 | 36.56 | 0.08 |
| Std. Dev. | 4.42 | 8726.67 | 7.49 | 0.03 |

Source: Authors' calculation based on SWIID, WDI database, TED.

**Table A2.** List of countries and their income groups.

| **Country** | **ISO3** | **Income Group** |
|---|---|---|
| Armenia | ARM | 2000–2001—low income<br>2002–2016—low-middle income<br>2017–2020—upper-middle income |
| Belarus | BLR | 2000–2006—low-middle income<br>2007–2020—upper-middle income |
| Estonia | EST | 2000–2005—upper-middle income<br>2006–2020—high income |
| Georgia | GEO | 2000–2002—low income<br>2003–2014? 2016–2017—low-middle income<br>2015, 2018–2020—upper-middle income |
| Kazakhstan | KAZ | 2000–2005—low-middle income<br>2006–2020—upper-middle income |
| Kyrgyz Republic | KGZ | 2000–2012—low income<br>2013–2020—low-middle income |
| Latvia | LTU | 2000–low-middle income<br>2001–2008, 2010—upper-middle income<br>2009, 2012–2020—high income |
| Lithuania | LVA | 2000—low-middle income<br>2001–2008, 2010–2011—upper-middle income<br>2009, 2012–2020—high income |
| Moldova | MDA | 2000–2004—low income<br>2005–2019—low-middle income<br>2020—upper-middle income |
| Russian Federation | RUS | 2000–2003—low-middle income<br>2004–2011, 2015–2020—upper-middle income<br>2012–2014—high income |
| Ukraine | UKR | 2000–2001—low income<br>2002–2020—low-middle income |

Source: Compiled by authors based on the WDI database.

## Notes

[1]  Alpha-3 country codes are used here and below (see Appendix A).

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
