# Peer review of "The Labour Share, Government Expenditure and Income Inequality of Post-Soviet Countries"

_economies, doi:10.3390/economies11120288_

Round 1

Reviewer 1 Report

Comments and Suggestions for Authors

The paper explores the relationship between income inequality, measured by means of the Gini index, level of income and the labour share in sample of post-Soviet countries over the period 2000-2020. After having addressed some issues related to the possible endogeneity in the determination of the level of income, the author(s) finds that the labour share (the level of income) improves (deteriorates) the fairness of the distribution of income except of high-income countries. In addition, the author(s) claims that mediating role of publica expenditures on such a relationship is mediated by the degree of democracy observed within the sample of countries.

The paper is quite interesting because the extend of inequality is related both to economic and institutional factors. As far as I can see, however, the paper suffers for some drawbacks that should be considered seriously. First, in the introduction as well as in the literature review, there is some confusion between the distribution of income and the distribution of wealth. The two issues are related but they remain different along several dimensions. Second, the author(s) does not address the possible endogeneity of labour share. Moreover, the claimed results appear as the outcome of some non-linearities between the involved variables. Nevertheless, the author(s) consider only linear relationships.

Comments on the Quality of English Language

The paper is awkwardly written – including the abstract – and it should be completely re-edited with the help a native English speaker.

Author Response

We would like to thank the reviewer for taking the time to read our work and for providing valuable feedback. We have addressed all the comments. Please, see the answers in the attached file.

Reviewer 2 Report

Comments and Suggestions for Authors

An interesting in subject and method paper. Some comments for the overall improvement:

1. Abstract: Add the data time spam - 2000-2020 or 20 y. e.t.c

2. Justify your model in several manners. Specifically:

2a  Define subscript t and its values

2b Defend your choice of Log transformation. How this improved the model fitting? Did you considered the other members of the power transformation family?

2c Present the final model(s) assumptions checking results and assessment.

3. Consider some of the latest publication on the subject and add them in the reference section

Comments on the Quality of English Language

Use of "the" in many cases all over the paper is missing

Long sentences have to be avoided (i.e. 48-53)

Use simple English (not direct translation from an other language

A review from a native English speaker is recommended

Author Response

We would like to thank the reviewer for taking the time to read our work and for providing valuable feedback. We have addressed all the comments. Please, see the answers in the table below.

1. Abstract: Add the data time spam - 2000-2020 or 20 y. e.t.c

We re-wrote the abstract and added the time spam

2. Justify your model in several manners. Specifically:

2a  Define subscript t and its values

2b Defend your choice of Log transformation. How this improved the model fitting? Did you considered the other members of the power transformation family?

2c Present the final model(s) assumptions checking results and assessment.

We added several comments to the model description on page 11

3. Consider some of the latest publication on the subject and add them in the reference section

The reference section includes many recent publications on the subject

Reviewer 3 Report

Comments and Suggestions for Authors

An interesting study, with some methodological issues to consider.

The introduction provides a reasonable overview of the study's purpose, research questions, and its relevance. However, it could benefit from more detail on the specific context and background of the study. This information may help readers better understand the significance of the study's findings and contextualize its implications. Additionally, the introduction could benefit from a clearer statement of the study's hypotheses or research questions to guide readers through the study's objectives.

The method section provides a fair description of the study's design, data collection, and analysis procedures. Additionally, while the researchers describe their analytical approach, they do not provide a detailed explanation of their statistical methods, which may impinge on the readers' ability to assess the validity of the results.

The findings section provides a good presentation of the study's results including their significance. However, it could benefit from more discussion of the limitations of the study and potential alternative explanations for the observed results. The researchers do not discuss in detail the potential confounding variables that may have influenced the analysis, which may limit readers' ability to assess the validity of the study's results.

For instance, numbers presented for the Baltic republics may give reason to believe that membership in the EU may have an undue influence on some of the input data (e.g. GDP). It therefore seems that the heterogeneity of economic development between these 11 former Soviet countries may not have been sufficiently accounted for.

Given the serial dependence of the data due to the usage af annual information, the issue of potential autocorrelation is not sufficiently explored. It is not clear that the conveyed relationship in Figure 8 is realistic.

Discounting for data points of productivity < 10,000, it seems the relationship between LSH and productivity is actually the other way around. The lack of a consistent trend between the results of the analysis of the 5 percentile models (as per Table 1) is not sufficiently elaborated.

Some odd referencing, for instance 'as Piketty argued in 2014 (Piketty 2014)'.

Comments on the Quality of English Language

 Minor editing of English language required

Author Response

We would like to thank the reviewer for taking the time to read our work and for providing valuable feedback. We have addressed all the comments. Please, see the answers in the attached file

Reviewer 4 Report

Comments and Suggestions for Authors

The authors proposed an interesting study, "The labour share, government expenditures and income inequality in post-Soviet countries." The paper is well-structured and conveys a deal of information. I want to suggest a few suggestions to improve the manuscript's quality and readability.

1.       If the authors use abbreviations, they must be used in a systematic way (e.g., GDP,  USSR,….., etc).

2.      The major defect of this study is the debate or argument is not clearly stated in the introduction session. Hence, the contribution is weak in this manuscript. I suggest the authors enhance the contribution part.

3.      The theoretical explanation section is missing. Kindly add a theoretical part with concrete and cohesive arguments regarding the studied variables.

4.      The findings should be compared and contrasted with prior findings for support and further insight and analysis. Also, the findings should be explained from the perspective of the study market.

5.      The policy formulation part is missing in the study. Please suggest some concrete and relevant policy implications based on the obtained results.

Author Response

We would like to thank the reviewer for taking the time to read our work and for providing valuable feedback. We have addressed all the comments. Please, see the answers in the table below.

If the authors use abbreviations, they must be used in a systematic way (e.g., GDP,  USSR,….., etc).

We have reviewed the use of abbreviations and made corrections.

The major defect of this study is the debate or argument is not clearly stated in the introduction session. Hence, the contribution is weak in this manuscript. I suggest the authors enhance the contribution part.

We expanded Introduction section and explained main hypothesis of the study.

 The theoretical explanation section is missing. Kindly add a theoretical part with concrete and cohesive arguments regarding the studied variables.

We provide theoretical background for our study in the literature review section

 The policy formulation part is missing in the study. Please suggest some concrete and relevant policy implications based on the obtained results.

We added policy applications  to the conclusion part

Round 2

Reviewer 1 Report

Comments and Suggestions for Authors

Accept

Author Response

Thank you very much the recognition of our work

Reviewer 3 Report

Comments and Suggestions for Authors

This looks now quite improved.

I would however still suggest the authors re-consider their interpreatation of data displayed in figure 8.

From my perspective this graph suggests two different relationships between productivity and LSH:

< 15,000 productivity (negative relationship) and

> 15,000 productivity (positive relationship).

The statistical significance of the difference between coefficients could be shown via Chow regression.  

Comments on the Quality of English Language

none

Author Response

Thank you for your comments and recognition of our work. We appreciate it.
We have tested Chow regression and added the results after Fig.8 We have also redesigned the Fig.8

Reviewer 4 Report

Comments and Suggestions for Authors

Accept.

Author Response

Thank you very much for the recognition of our work.